# Transcriptome Analysis for Genes Associated with Small Ruminant Lentiviruses Infection in Goats of Carpathian Breed

**DOI:** 10.3390/v13102054

**Published:** 2021-10-13

**Authors:** Monika Olech, Katarzyna Ropka-Molik, Tomasz Szmatoła, Katarzyna Piórkowska, Jacek Kuźmak

**Affiliations:** 1Department of Biochemistry, National Veterinary Research Institute, 24-100 Pulawy, Poland; jkuzmak@piwet.pulawy.pl; 2Department of Animal Molecular Biology, National Research Institute of Animal Production, Krakowska 1, 32-083 Kraków, Poland; katarzyna.ropka@izoo.krakow.pl (K.R.-M.); tomasz.szmatola@izoo.krakow.pl (T.S.); katarzyna.piorkowska@izoo.krakow.pl (K.P.); 3Center for Experimental and Innovative Medicine, University of Agriculture in Krakow, Rędzina 1c, 30-248 Kraków, Poland

**Keywords:** small ruminant lentiviruses, SRLV, RNA-seq, NGS sequencing, proviral load, goat

## Abstract

Small ruminant lentiviruses (SRLV) are economically important viral pathogens of sheep and goats. SRLV infection may interfere in the innate and adaptive immunity of the host, and genes associated with resistance or susceptibility to infection with SRLV have not been fully recognized. The presence of animals with relatively high and low proviral load suggests that some host factors are involved in the control of virus replication. To better understand the role of the genes involved in the host response to SRLV infection, RNA sequencing (RNA-seq) method was used to compare whole gene expression profiles in goats carrying both a high (HPL) and low (LPL) proviral load of SRLV and uninfected animals. Data enabled the identification of 1130 significant differentially expressed genes (DEGs) between control and LPL groups: 411 between control and HPL groups and 1434 DEGs between HPL and LPL groups. DEGs detected between the control group and groups with a proviral load were found to be significantly enriched in several gene ontology (GO) terms, including an integral component of membrane, extracellular region, response to growth factor, inflammatory and innate immune response, transmembrane signaling receptor activity, myeloid differentiation primary response gene 88 (MyD88)-dependent toll-like receptor signaling pathway as well as regulation of cytokine secretion. Our results also demonstrated significant deregulation of selected pathways in response to viral infection. The presence of SRLV proviral load in blood resulted in the modification of gene expression belonging to the toll-like receptor signaling pathway, the tumor necrosis factor (TNF) signaling pathway, the cytokine-cytokine receptor interaction, the phagosome, the Ras signaling pathway, the phosphatidylinositol 3-kinase (PI3K)/protein kinase B (AKT) (PI3K-Akt) signaling pathway and rheumatoid arthritis. It is worth mentioning that the most predominant in all pathways were genes represented by toll-like receptors, tubulins, growth factors as well as interferon gamma receptors. DEGs detected between LPL and HPL groups were found to have significantly enriched regulation of signaling receptor activity, the response to toxic substances, nicotinamide adenine dinucleotide (NADH) dehydrogenase complex assembly, cytokine production, vesicle, and vacuole organization. In turn, the Kyoto Encyclopedia of Genes and Genomes (KEGG) pathway tool classified DEGs that enrich molecular processes such as B and T-cell receptor signaling pathways, natural killer cell-mediated cytotoxicity, Fc gamma R-mediated phagocytosis, toll-like receptor signaling pathways, TNF, mammalian target of rapamycin (mTOR) signaling and forkhead box O (Foxo) signaling pathways, etc. Our data indicate that changes in SRLV proviral load induced altered expression of genes related to different biological processes such as immune response, inflammation, cell locomotion, and cytokine production. These findings provide significant insights into defense mechanisms against SRLV infection. Furthermore, these data can be useful to develop strategies against SRLV infection by selection of animals with reduced SRLV proviral concentration that may lead to a reduction in the spread of the virus.

## 1. Introduction

The maedi visna virus (MVV) and the caprine arthritis encephalitis virus (CAEV) belong to the group called small ruminant lentiviruses (SRLV) within the *Retroviridae* family. Molecular epidemiology studies revealed that both viruses represent a broad spectrum of genetic variants that can infect sheep and goats. To date, four main genotypes have been described, but molecular information on new subtypes is continuously updated, showing a high genetic and antigenic heterogeneity [1]. Infections with SRLV, which are spread worldwide, cause multi-organ failure usually over a long period of time and can lead to severe diseases such as pneumonia, mastitis, arthritis, wasting, and encephalitis [2]. Moreover, they contribute to economic losses in small ruminant production and affect animal welfare deterioration [3,4].

There is no effective vaccine or treatment preventing animals from SRLV infection. Several practices for controlling or preventing SRLV infection have been developed, such as serological testing with culling or segregation of infected animals, replacement of infected animals with offspring from seronegative mothers, or artificial rearing of newborn animals separated from the infected mothers immediately after birth [5,6]. These practices can be effective when carefully designed and applied continuously to eradicate the progression of the infection [6,7,8,9,10]. However, such an approach is often costly and time-consuming. The high genetic variability of SRLV and the absence of sensitive diagnostic tests that are able to detect all strains are additional challenges reducing the effective implementation of eradication programs.

The dynamics of the host immune response to SRLV infection are still not fully understood. Several attempts to identify host factors associated with resistance to SRLV infections have been made, and some loci were identified [11,12,13,14,15]. In particular *TMEM154* gene (transmembrane protein 154)*, TMEM38A* (transmembrane protein 38A), *CCR5* (chemokine(C-C motif) receptor type 5), *MHC* (major histocompatibility complex), *ZNF389* (zinc-finger protein 389), *TLRs* (toll-like receptors), *APOBEC3* (apolipoprotein B editing complex 3), *TRIM5* (tripartite motif protein 5 alpha), *Tetherin/BST-2* (bone marrow stromal cell antigen 2) and other cytokines (interleukin 2 (IL2), interleukin 2 receptor (IL2R), tumor necrosis factor alfa (TNF-α), interleukin 4 (IL4), interleukin 8 (IL8), interleukin 6 (IL6), interleukin 16 (IL-16), interferon gamma (IFN-γ), transforming growth factor beta (TGF-β1), monocyte chemoattractant protein-1 (MCP-1), granulocyte-macrophage colony-stimulating factor (GM-CSF)) and chemokine ligands (C-C motif ligand 2 (CCL2), C-C motif ligand 5 (CCL5), C-C motif ligand 20 (CCL20)) seem to have an important role in the SRLV infection susceptibility/resistance [12,13,15,16,17,18,19,20,21,22,23,24,25,26,27,28,29,30,31,32,33]. However, additional studies are still necessary to learn more about genes involved in SRLV immunity.

In many persistent viral infections, viral load is reported to estimate the likelihood of pathogenesis and disease progression. For retroviruses, including lentiviruses, in which genomes are integrated with the host genome, proviral load (PL) is a risk factor determining disease prediction [34]. It was shown that animals with high PL showed more tissue lesion severity, indicating that proviral concentration is positively correlated with the presence and severity of clinical disease symptoms [35,36]. Elimination of animals predisposed to high PL can limit the outcome of clinical signs and spread of the virus since these animals are also highly efficient in shedding the virus [37]. On the contrary, some studies indicated potential restriction in low PL carriers and referred to them as long-term non-progressors. These animals showed competent antibody response in the absence of productive virus replication leading to minimizing the spread of the virus within the flock [38]. Consistent with this, approaching genetics factors associated with low PL in animals infected with SRLV could be used to control SRLV infection, especially in flocks with a high level of seroprevalence.

In this study, RNA sequencing (RNA-seq) was used to identify genes associated with high (HPL) and low (LPL) proviral load in goats of Carpathian breed naturally infected with SRLV. The results provide unique insights for further exploration and understanding patterns of the host responses to SRLV infection in goats. A deep understanding of transcriptome profile changes may provide additional information on the contribution of specific genes responsible for the course of infection with SRLV. Additionally, these data can be useful to develop strategies against SRLV infection by elimination and/or selection of animals with reduced SRLV provirus concentration, which may lead to limitation of viral spread and can improve the welfare of animals and prolong their life.

## 2. Methods

### 2.1. Animals and Blood Sample Collection

Whole blood was taken from 27 adult goats by jugular venipuncture and stabilized in ethylenediaminetetraacetic acid (EDTA) and Tempus blood RNA tubes. Goats represented the Carpathian breed, and they were owned by the National Research Institute of Animal Production in Krakow. All goats were healthy and maintained in one flock in the same environment. This involved being housed indoors in sheds, aside from during the grazing season (April–November), where they spent daily hours in pastures outdoors. It also involved the feeding conditions (summer feeding based on pasture or green fodder and winter feeding consisting mainly of hay and oats). Serological status of animals for SRLV infection was confirmed by enzyme-linked immunosorbent assay (ELISA) (ID Screen MVV/CAEV Indirect Screening test, IDVet, Grabels, France) according to the manufacturer’s recommendations. Blood samples were collected from all animals on the same day. At the time when blood was taken, none of the goats exhibited any clinical signs of the disease. All procedures associated with animal handling and treatments were approved (no 37/2016) by the Local Ethical Committee on Animal Testing at the University of Life Sciences in Lublin (Poland).

### 2.2. Proviral Load Quantification

DNA was extracted from peripheral blood leukocytes (PBLs), and proviral DNA was quantified by the real-time polymerase chain reaction (PCR) using Rotor-Gene Q Series ver. 2.0.3 (Qiagen, Hilden, Germany) with primers and probe specifically designed for SRLV A5 subtype, which circulation in this flock was previously confirmed [39]. Sequences of forward and reverse primers and probe were CA5F (5′ TGGGAGTAGGACAAACAAATCA 3′), CA5R (5′ TGACATAT GCCTTACTGCTCTC 3′) and CA5P (5′ 6-FAM-TCACCCATTGTAGGCATAGCTGCC-BHQ-1 3′), respectively. A reference plasmid encompassing the target *gag* region was generated by the cloning of a 625 bp fragment into pDrive plasmid used to generate a standard curve based on 10-fold serial dilutions of plasmid DNA from 10^8^ to 1^0^. Amplification was performed in a total volume of 20 μL, according to the following cycling conditions: initial incubation and polymerase activation at 95 °C for 15 min and followed by 45 cycles of 94 °C for 60 s and 60 °C for 60 s. The reaction mixture for each PCR test contained 10 μL 2× QuantiTect Multiplex NoROX PCR buffer (Qiagen, Hilden, Germany), 400 nM of each of the primers, 200 nM of the specific probe, 5 μL of extracted genomic DNA. A non-template control (diethylpyrocarbonate (DEPC) H_2_O) was included in each run. All samples were tested in duplicate, and the results were expressed as a mean copy number of provirus per 500 ng of genomic DNA of each goat. Then, these data were used for further statistical analysis, thereby allowing the identification of goats with a high (HPL) and low (LPL) proviral load.

### 2.3. Transcriptome Sequencing and Data Analysis

The total RNA was isolated from the whole blood of goats using MagMAX™ for Stabilized Blood Tubes RNA Isolation Kit (Thermo Fisher Scientific, Waltham, MA, USA ), according to the protocol. The possible RNA contamination with DNA was removed using TURBO DNase™ (Thermo Fisher Scientific, Waltham, MA, USA). The quality and quantity of obtained RNA were checked using the Nanodrop 2000 spectrophotometer (Thermo Scientific, Waltham, MA, USA) and TapeStation 2200 System (Agilent, Santa Clara, CA, USA) using Agilent RNA ScreenTape (Agilent, Santa Clara, CA, USA). The samples with an RIN (RNA integrity number) value above 8 were used for further analysis. The cDNA libraries were prepared from 300 ng of total RNA with the use of the TruSeq RNA Kit v2 kit protocol (Illumina, San Diego, CA, USA). Each library was ligated with adaptors with different index barcodes. The quality and quantity of libraries were assessed using Qubit 2.0 fluorometer (Invitrogen, Carlsbad, CA, USA ) and TapeStation 2200 (Agilent, Santa Clara, CA, USA) with D1000 ScreenTape (Agilent, Santa Clara, CA, USA). Libraries were pooled and sequenced by synthesis, using HiSeq High-Output v4-SR (Illumina, San Diego, CA, USA) into 50 single-end cycles, according to the protocol. The quality of the reads was assessed with FastQC software [40]. Then, we used Flexbar software [41] to remove adapters, reads shorter than 35 base pairs, and those with a phred quality score lower than 30. Processed reads were mapped to the goat reference genome *Capra hircus* ARS1 (GCA_001704415.1) with Tophat software [42] on default parameters. Next, the mapped reads were counted into Ensembl GTF version 97 annotation intervals using HTSeq-count software [43]. Differential expressed genes (DEG) were estimated using DESeq2 software [44] with default parameters. Genes with *p*-adjusted < 0.05 (Benjamini–Hochberg *p*-value adjustment) and fold change >1.3 were regarded as differentially expressed and included in further annotation analysis.

### 2.4. GO Enrichment and Pathways Analysis

The gene ontology analysis (GO) and pathways analyses were performed on all significant differentially expressed genes (DEGs) sets. The gene set enrichment analysis (GSEA) was performed with the use of WebGestalt software with Fisher’s exact test (http://webgestalt.org/ accessed on 31 July 2021). For pathway functional analysis, David software v6.8 (Fisher Exact test) and the Kyoto Encyclopedia of Genes and Genomes (KEGG) with KEGG mapper search pathway tools were applied. The significantly enriched pathways were identified based on the *p*-values obtained from a Fisher exact test [45]. The latest version of the *Capra hircus* ARS1 (GCF_001704415.1) reference genome was used.

### 2.5. Quantitative Polymerase Chain Reaction (qPCR) Analysis

Validation of the RNA-seq results was carried out using the real-time PCR method for nine DEGs (Appendix A Appendix A), selected based on their important function in viral infection and/or previous reports confirming their association with proviral concentration. The qPCR was performed for the validation of RNA-seq results of all samples analyzed using the RNA-seq method (for validation), as well as for all samples tested by qPCR, which was tasked with the estimation of the correlation between PL (SRLV copy number) and gene expression levels (for a correlation with SRLV copy number). The cDNA was prepared from 250 ng of total RNA using a high-capacity RNA-to-cDNA Kit (Thermo Fisher Scientific, Waltham, MA, USA) according to protocol. The transcript level of selected genes was estimated on QuantStudio 7 Flex (Applied Biosystems, Thermo Fisher Scientific, Waltham, MA, USA), and for each gene, the reaction was carried out in three replications using Sensitive RT HS-PCR Mix EvaGreen (A&A Biotechnology, Gdynia, Poland). The expression was calculated using the delta-delta CT method according to Pfaff [46] and based on *HPRT1* and *ACTB* reference controls [47]. The comparison between next-generation sequencing (NGS) data (RNA-seq) and relative quantity obtained by real-time PCR method was performed using Spearman correlation with the use of R software [48].

### 2.6. Statistical Analysis

To classify the animals into HPL and LPL groups, a copy number calculated by qPCR per each animal was used as a potential cut-off value, and a Box–Cox transformation was used to achieve normal distribution. The t-Student test was employed to determine the significance of the differences between the two potential groups of animals. Finally, the cut-off value was chosen based on the lowest *p*-value to distinguish between the HPL and LPL groups, which was additionally confirmed by the Welch test (*p* < 0.0001).

The phenotypic and physiological variables between HPL and LPL goats were analyzed using TIBCO Software Inc. Statistica (Data Analysis Software System, Palo Alto, CA, USA), version 13 (2017). The association between physiological differences (stage of lactation, mastitis and other diseases, abortion) between HPL and LPL goats were tested by a chi-square test with appropriate correction. The variables included age, parity number, body weight, and milk yield (kg) and were analyzed with Pearson correlation. All variables were also tested by logistic regression models. The normality of the distribution and the homogeneity of variance was tested by Shapiro–Wilk and Brown–Forsyth tests, respectively. Differences were considered significant when *p* > 0.05.

## 3. Results

### 3.1. Classification of Goats on HPL and LPL

A total of 24 animals from the flock tested in this study were seropositive by ELISA and positive to quantitative polymerase chain reaction (qPCR), whereby confirming the infection of SRLV. Three goats were negative in both ELISA and qPCR. The average number of proviral copies varied from 1 to 106 per 500 ng of genomic DNA, and these values showed skewed distribution with a relatively limited number of animals with a high concentration of provirus. Animals were classified into HPL (mean ± SD; 82.39 ± 13.14) and LPL group (mean ± SD; 14.31 ± 14.23) (Figure 1).

In addition, the statistical analyses of the phenotypic and physiological differences between HPL and LPL goats were performed. Variables analyzed included age, parity number, body weight, milk yield, stage of lactation, disease occurrence (including mastitis), and abortion. A moderate correlation between proviral load and age (r = 0.35, *p* = 0.035), as well as parity number (r = 0.38, *p* = 0.024), was observed; however, no significant differences between HPL and LPL goats were noted when other variables were analyzed. Finally, eight goats (four goats with HPL and four goats with LPL) that were phenotypically and physiologically the most homogeneous were then carefully selected for whole blood transcriptome analysis. All these animals were female, unrelated within biological groups, clinically healthy, and multiparous after parturition. Their average age was 7.25 ± 1.98 years, the average body weight was 45.3 ± 5.15 kg, and they produced on average 266.6 ± 96.0 kg of milk per year.

Moreover, partial (*gag* and LTR) sequences of the SRLV genome were analyzed, and no significant mutations/differences between sequences obtained from HPL and LPL goats that could alter proviral load were observed.

### 3.2. RNA-seq Data

#### 3.2.1. Transcriptome Quantification

Transcriptome analysis was performed using whole blood taken from 11 selected goats. These animals included four goats with high (HPL group) and four goats with low (LPL group) SRLV proviral load, as well as three uninfected goats (control group). After next-generation sequencing (NGS) and data filtration, the average number of reads obtained per sample was about 34.3 mln. On average, 83.8% of reads were mapped to the reference *Capra hircus* genome (GCA_001704415.1) (Appendix A Appendix A). Furthermore, the principal component analysis (PCA) showed that the analyzed groups of animals formed distinct clusters, which confirmed the presence of two groups of goats with high and low proviral concentrations (Appendix A Appendix A).

#### 3.2.2. DEGs Analysis

The whole blood transcriptome sequencing using the NGS approach allowed the identification of 1130 DEGs between control and LPL groups, 411 between control and HPL groups, and 1434 significant DEGs between HPL and LPL groups (Figure 2).

When the set of DEGs between uninfected and both HPL and LPL goats was compared, the 1046 genes were detected. Among this DEGs panel, 408 genes were identified as downregulated in both HPL and LPL groups, while 638 genes were upregulated. This gene set was used for subsequent analysis as being potentially associated with immune response to SRLV infection. Among all 1434 DEGs identified between LPL and HPL goats, 571 were upregulated and 863 downregulated.

In the panel of 1046 DEGs differentiated uninfected animals from those with SRLV proviral load, the genes with the highest expression changes and downregulated in animals with proviral load were *KITLG* (KIT ligand—mast cell growth factor) 256-fold change; *HHPI* (hedgehog interacting protein) 234-fold change; *SLC17A6* (vesicular glutamate transporter 2) 222-fold change and *P2RX2* (purinergic receptor P2X 2) 186-fold change. In turn, the most upregulated genes in the blood of goats with proviral load were *PDGFRB* (platelet-derived growth factor receptor beta) 76-fold change; *TUBA4A* (tubulin alpha 4a) 67-fold change; *TNNI3* (troponin I3, cardiac type) 66 FC and *TMEM176B* (transmembrane protein 176B) 59FC.

Moreover, the following family of genes in which expression was deregulated in response to SRLV infection were identified: the zinc-finger gene family (*ZNF*; 8 genes); a transmembrane protein (*TMEM*; 10 genes); the solute carrier family genes (*SLC*; 14 genes); signaling protein (11 genes), toll-like receptors (*TLR*; 4 genes), and tubulins (*TUBB*; 4 genes).

When the groups of LPL vs. HPL were compared, the highest differences in gene expressions were detected for downregulated genes for which the decrease in transcript-level abundance reached up to 419-fold change (FC) for solute carrier family 22 member 1 (*SLC22A1*) gene. The most significant deregulated genes showing the highest FC in gene expression between analyzed groups are presented in Figure 3 and Appendix A Appendix A. The group of genes for which expression was deregulated in response to the provirus concentration were: the zinc-finger gene family (*ZNF*; 23 genes); a transmembrane protein (*TMEM*; 16 genes); the solute carrier family genes (*SLC*; 22 genes); the *NADH*: ubiquinone oxidoreductase supernumerary subunits (*NDUF*) genes (20 genes); ATP synthase genes (15 genes); interleukin and interleukin receptors (12 genes); the sorting nexin family (*SNX*; 5 genes) and the translocase of inner mitochondrial membrane family (*TIMM*, 5 genes).

### 3.3. Gene Ontology and Pathways Annotation

#### 3.3.1. DEGs Detected between Control Group and Groups with Proviral Load

The gene ontology analysis showed the significant enrichment of several GO terms (Table 1). The most overrepresented were integral components of the membrane, which involved 13 DEGs, 6 extracellular region genes, and 5 response to growth factor genes. Other detected GO terms related to inflammation and innate immunity were represented by four genes belonging to the toll-like receptors family, *TLR2*, *TLR4*, *TLR8*, and *TLR7*, which were all significantly upregulated in the infected goats (Figure 4).

Our results also showed significant deregulation of selected pathways in response to viral infection (Table 2). The presence of SRLV proviral load in blood cells resulted in modification of expression in genes belonging to toll-like receptor signaling pathway (FDR > 0.05); TNF signaling pathway (FDR > 0.01); Cytokine-cytokine receptor interaction (FDR > 0.04) and phagosome (FDR > 0.02) (Figure 5). It is worth mentioning that the most predominant genes in all pathways were the genes represented by toll-like receptors, tubulins, growth factors, as well as interferon gamma receptors. The highest number of downregulated genes were detected within the Ras signaling pathway. These pathways allowed the identification of *PLA2G1B* (phospholipase A2 group IB) and *KITLG* (KIT ligand) DEGs, and both were considered as strongly related to viral infection.

#### 3.3.2. DEGs Detected between LPL and HPL Groups

GO enrichment analysis allowed the detection of the most represented GO term, 10 up- and 10 downregulated (Figure 6).

In the group of downregulated genes, most of them were associated with: regulation of signaling receptor activity (*p*-value < 0.0001) (53 genes, e.g., chemokine (C-C motif), ligand 2 (*CCL2*), chemokine (C-X-C motif), ligand 5 (*CXCL5*), TNF superfamily member 11 (*TNFSF11*), C-C motif chemokine ligand 17 (*CCL17*), C-X-C motif chemokine ligand 9 (*CXCL9*), macrophage migration inhibitory factor (*MIF*)); the response to toxic substances (43 genes, e.g., hemoglobin subunit mu (*HBM*), cholinergic receptor nicotinic beta 2 subunit (*CHRNB2*), LY6/PLAUR domain containing 1 (*LYPD1*), gonadotropin-releasing hormone 1 (*GNRH1*)), and NADH dehydrogenase complex assembly (21 genes, e.g., NADH: ubiquinone oxidoreductase subunit A13 (*NDUFA13*), NADH: ubiquinone oxidoreductase subunit S5 (*NDUFS5*), NADH: ubiquinone oxidoreductase core subunit S7 (*NDUFS7*), NADH: ubiquinone oxidoreductase subunit B9 (*NDUFB9*)). The specific genes belonging to detected GO were presented in Table 3 and in Appendix A Appendix A. In the HPL group, an increased expression for 95 genes was identified (e.g., ADAM metallopeptidase with thrombospondin type 1 motif 3 (*ADAMTS3*), interleukin 15 (*IL15*), chemerin chemokine-like receptor 1 (*CMKLR1*), nucleotide-binding oligomerization domain containing 2 (*NOD2*), C-C motif chemokine receptor 2 (*CCR2*), interleukin 6 receptor (*IL6R*), interleukin 1 alpha (*IL1A*)) that represented GO term cytokine production (FDR < 0.0001) (Table 4, Appendix A Appendix A). In addition, a high number of upregulated genes related to vesicle organization was detected (38 genes; FDR < 0.030), as well as vacuole organization (34 genes, FDR < 0.035).

To better show the interaction between the broad set of genes represented by the cytokine production GO term, the gene network was prepared according to String software. The analysis indicated that some of the identified DEGs were involved in multiple biological processes related to positive and negative regulation of cytokine production and control in response to stimulus in both adaptive and innate immunity (Figure 6). Such an approach allowed to pinpointed the upregulated genes with the highest number of interactions (Table 5).

Identified DEGs were also analyzed for their involvement in biological pathways. Thus, genes have been assigned to pathways involved in the acquired or antigen-specific immune response (B-cell receptor and T-cell receptor signaling pathways; natural killer cell-mediated cytotoxicity and Fc gamma R-mediated phagocytosis). Moreover, DEGs belonged to the pathways responsible for recognition of pathogen, signal transduction, and early immune responses: toll-like receptor signaling pathway; tumor necrosis factor (TNF) signaling pathway; mammalian target of rapamycin (mTOR) signaling; and forkhead box O (Foxo) signaling pathway. The comparison of the whole blood transcriptome of goats with different provirus copy numbers allowed the detection of the Ras signaling pathway (17 DEGs), inflammatory mediator regulation of transient receptor potential (TRP) channels (12 DEGs), and hypoxia-inducible factor 1 (HIF-1) signaling pathway (12 DEGs), which are considered as critical to control cytokine production, cell differentiation, function, and cytotoxicity. The panel of genes was identified (paxillin (*PXN*); profilin 1(*PFN1*); actin-related protein 2/3 complex subunit 5 (*ARPC5*); cytoplasmic FMR1 interacting protein 1 (*CYFIP1*); IQ motif containing GTPase activating protein 1 (*IQGAP1*)), which also represent regulation of the actin cytoskeleton. The genes belonging to the selected pathways are presented in Table 6.

### 3.4. qPCR Results

DEGs from different functional groups, including the following genes: *CCL2*, *CXCL5*, *IL15*, C-X-C motif chemokine receptor 3 (*CXCR3*), *MIF*, *NOD2*, *CCR*, B-cell lymphoma 2 (*BCL2*), and IL-2-inducible T-cell kinase (*ITK*), were selected for further validation by qRT-PCR. This analysis revealed an agreement with the RNA-seq results: a high and significant correlation was detected for *IL15*, *CXCR3*, and *NOD2* genes (Table 7). For other genes, the correlation was not significant, which may be related to genome annotation still being under development and continued limited knowledge of all spliced variants of the studied genes.

The correlation between provirus copy number and gene expression levels carried out using samples from all animals tested from the flock showed that selected DEGs as *CCL2* and *CXCL5* (*p*-value < 0.001), *CCR* and *BCL2* (*p*-value < 0.01), and *ITK* and *NOD2* genes (*p*-value < 0.05) were significantly associated with SLRV copy number.

## 4. Discussion

To better understand the role of genes involved in the host response to SRLV infection, the RNA-seq method was applied to compare the whole gene expression profile in uninfected goats with those carrying relatively high and low SRLV proviral loads. Data obtained in this study enabled us to identify 1130 DEGs between control and LPL groups, 411 between control and HPL groups, and 1434 significant genes showing changed expression levels depending on provirus copy number. Out of the panel of 1434 DEGs differentiated HPL and LPL goats, only 10 DEGs were shared between both the control vs. LPL and control vs. HPL groups. This indicates that proviral load might be the main driver and risk factor determining disease prediction in goats infected with SRLV [35]. Here, we focused on the analysis of some DEGs only being involved in immunological processes since both innate and adaptive immune responses are known to play a crucial role in controlling the course of SRLV infection.

It was shown that SRLV infection influences the expression of a cytokine network that plays a pivotal role in the activation of the immune system and SRLV-related pathogenesis [20]. Our findings indicated that 95 of the DEGs that were involved in multiple biological processes of cytokine production were overexpressed in HPL goats. These animals showed upregulated expression of interleukin 15 (IL-15) and interleukin 1 alpha (IL-1α) and receptors for IL-10 (*IL10Rβ*), IL-13 (*IL13Rα1*), IL-15 (*IL15Rα*), IL-2 (*IL2Rα*) and IL-4 (*IL4R*). This observation partly confirmed results obtained by Ravazzolo et al. [35], who did not find prominent differences in the expression of several interleukins in goats with different SRLV proviral loads. The level of IL-15 seems to be associated with the proviral load, as was also seen in patients infected with human immunodeficiency virus (HIV) with high viral load [49]. There is limited knowledge about the expression of IL-1α in the course of SRLV infection. Jarczak et al. [50] observed down-regulation of IL-1α mRNA in the blood of infected goats, suggesting that lentivirus infection may inhibit the expression of this gene. However, this fact was not confirmed in our study where IL-1α was upregulated in HPL goats. IL-1α is a proinflammatory cytokine that induces the expression of a variety of genes and synthesis of several proteins, which, in turn, induce acute and chronic inflammatory changes [51,52]. However, animals tested in this study did not show any clinical signs of infections, but we cannot exclude the presence of inflammatory processes, especially in goats with HPL, as the association between virus load and presence of inflammatory lesions was clearly evidenced [35,53].

Among the most interesting DEGs detected in this study and engaged in numerous biological processes of cytokine production were also toll-like receptor 2 (*TLR2*), toll-like receptor 4 (*TLR4*), toll-like receptor 6 (*TLR6*), cluster of differentiation 14 (*CD14*), and myeloid differentiation primary response gene 88 (*MyD88*), which are involved in TLR signaling. Toll-like receptors, a family of pattern recognition receptors (PRRs), are key elements of native immunity. While the role of SRLV-induced TLR signaling has not been widely studied in sheep and goats, it was shown that mutations in TLR7 and TLR8 may play an important role in susceptibility and/or resistance to SRLV infection [54,55]. Our results indicated that four genes belonging to the toll-like receptors family, *TLR2*, *TLR4*, *TLR8*, *TLR7*, were significantly upregulated in the infected goats compared to uninfected goats. However, we did not observe different expressions of *TLR7* and *TLR8* genes between goats with HPL and LPL. Only the genes encoding *TLR2*, *TLR4*, and *TLR6* were differentially expressed and were found to be upregulated in HPL groups. TLR2, TLR4, and heterodimers TLR2-TLR6 are TLR family members that have been involved in the recognition of viral structural and nonstructural proteins leading to inflammatory cytokine production [56,57]. The expression of CD14 (a co-receptor for the TLR4 and TLR2 response [58] and MyD88), an adaptor molecule that is critical for the signaling responses initiated through most TLRs, was also upregulated in HPL goats. We can conclude that immune response against SRLV is at least partially dependent upon TLR2 and TLR4 and correlated with the concentration of proviral DNA, as was shown in the HIV model based on the expression of TLR2 and TLR4 in monocytes [59].

Interferons (IFNs α, β, and γ) response is a highly robust and effective first line of defense against a wide variety of viral infections; however, results on IFNs expression during SRLV infection are contradictory [20,35,60,61]. When the IFN is synthesized, it binds to the interferon alpha receptor (IFNAR), the specific receptor for IFN-I on the cell membrane, formed by two subunits: IFNAR-1 and IFNAR-2. This binding activates the tyrosine kinases TYK-2 and JAK-1, leading to the activation of the JAK-STAT pathway, which is important in cytokine-mediated immune responses [62]. In our study, no differences in IFNs expression were observed between infected and uninfected goats, as well as between HPL and LPL goats. However, genes involved in IFN signaling, interferon regulatory factor 1 (*IRF1*), interferon receptor (*IFNAR1*), interferon-induced transmembrane protein 1 (*IFITM1*), interferon-inducible protein 1 (*IFIH1*), genes-encoded proteins of JAK/STAT family (*STAT2*, *STAT3*, *JAK2*, *JAK3*, *TYK2*) and interferon-induced protein with tetratricopeptide repeats (*IFIT1*, *IFIT2*, and *IFIT3*), were found to be one of the most upregulated genes in goats with HPL. Overexpression of these genes may result in higher activation of factors involved in antiviral responses. Our results may indicate that gene expression of INFs did not necessarily correspond with the protein concentration, which was also suggested by Jarczak et al. [50].

The zinc-finger (ZNF) proteins provide a particular interest in this analysis because 23 genes encoding these proteins were differentially expressed in HPL and LPL goats. Zinc-finger proteins have nucleic acid-binding domains that can serve to regulate multiple gene transcription. It has been established that a deletion variant near ZNF389 influenced SRLV proviral concentration in multiple sheep flocks [63]. The functional importance of ZNF during regulation of SRLV infection in goats is currently unknown, but our results strongly suggest that expression of these genes is dominant in response to the SRLV infection and can be associated with SRLV proviral concentration as was observed for HIV [64].

Another group of genes in which expression was dysregulated in response to the infection with SRLV and was associated with proviral concentration was the genes-encoded transmembrane proteins (TMEM; 16 genes). Recently, *TMEM154* and *TMEM38A* genes were identified as suitable candidates for SRLV resistance in sheep [15,16]. An amino acid substitution (E/K) at position 35 of the TMEM154 was associated with the lower concentration of SRLV provirus in sheep [18,19]. In the present study, we did not find any correlation between the expression of *TMEM154* and proviral load. However, our results provided evidence that other genes, such as *TMEM238*, *TMEM223*, *TMEM151*, *TMEM147*, *TMEM53*, were upregulated and may play an important role in the course of SRLV infection and provirus concentration.

## 5. Conclusions

In this study, we have demonstrated the changes in the transcriptome profile of goats showing high and low proviral load following infection with SRLV. A total of 1434 differentially expressed genes were involved in a variety of molecular and cellular defense mechanisms of immune response, cell cycle regulation, and cellular metabolism. Numerous genes have not been previously associated with lentiviral infection and may extend structural and/or regulatory networks implicated in the course of infection with SRLV (Appendix A Appendix A). The knowledge about these genes provides the basis for further work to identify genetic markers associated with SRLV infection and provirus concentration. Such markers may be used to eliminate animals predisposed to high proviral load and limit the outcome of clinical signs and spreading of the virus.

## Figures and Tables

**Figure 1 viruses-13-02054-f001:**
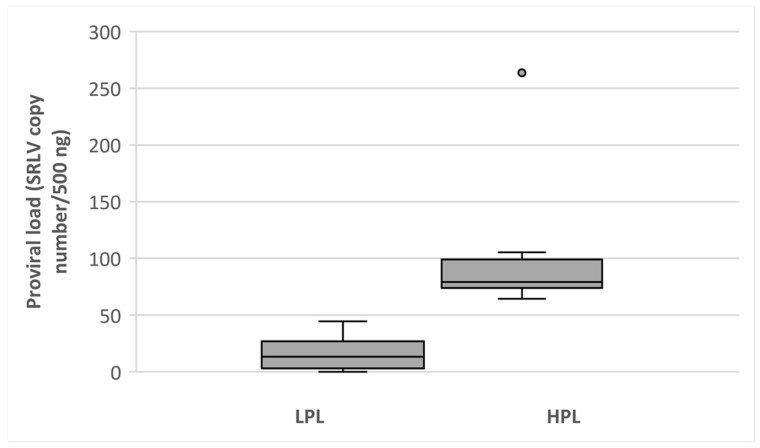
Proviral load distribution among goats with low (LPL) and high (HPL) proviral load. HPL—high proviral load, LPL—low proviral load.

**Figure 2 viruses-13-02054-f002:**
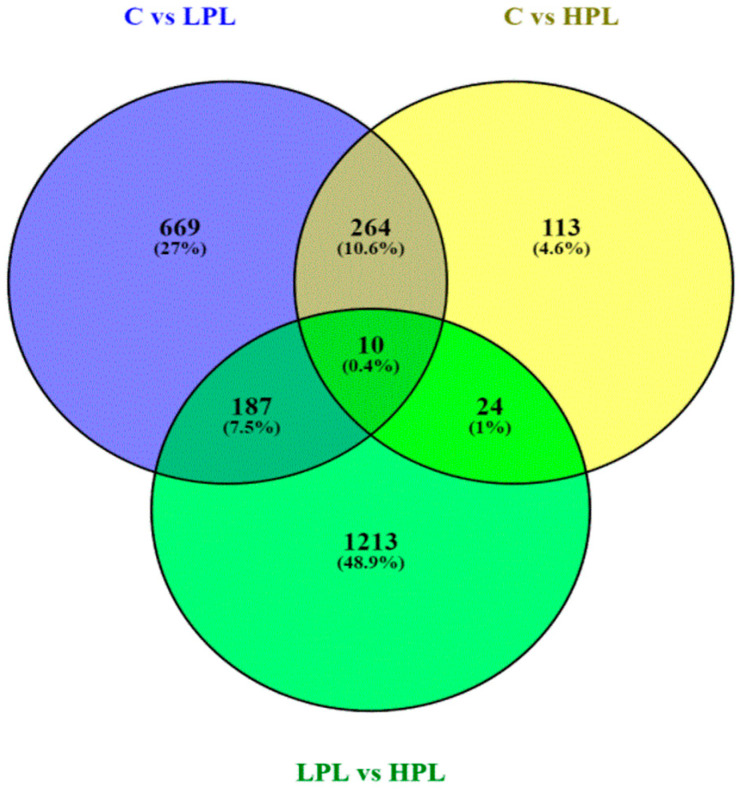
Venn diagram showing the number of overlapping DEGs between C vs. LPL, C vs. HPL, and LPL vs. HPL.

**Figure 3 viruses-13-02054-f003:**
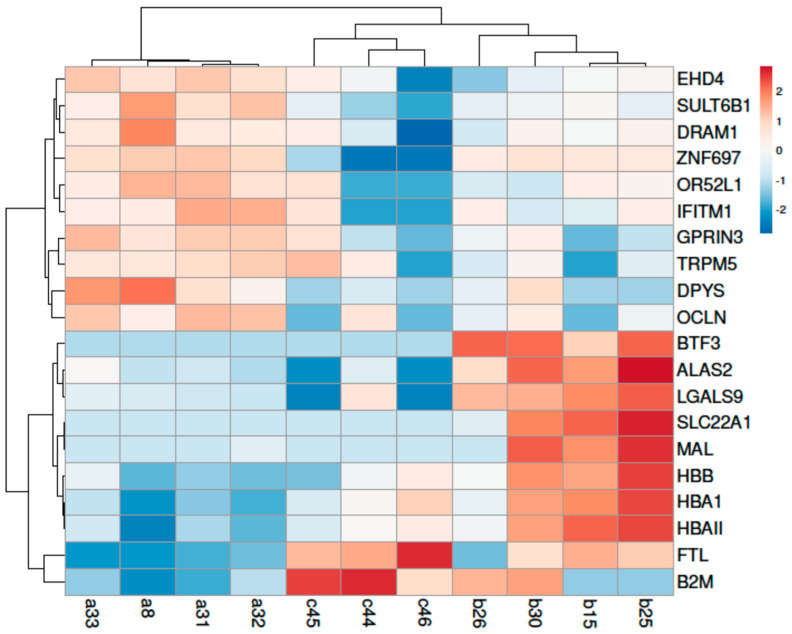
The heatmap of the most significantly deregulated genes between the uninfected goats and goats with high (HPL) and low (LPL) proviral loads. Animals with high (HPL) proviral load: b15, b25, b26 and b30; animals with low (LPL) proviral load: a8, a31, a32, and a33; Uninfected animals: c44, c45, and c46.

**Figure 4 viruses-13-02054-f004:**
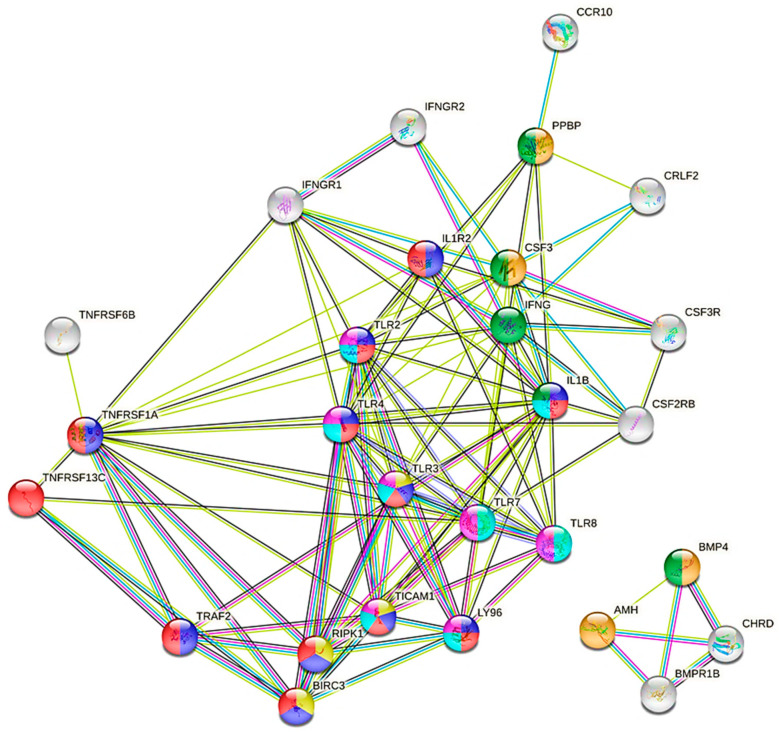
The interaction between differentially expressed genes involved in toll-like receptor signaling and cytokine-cytokine receptor interaction pathways (red—genes belonged to NF-kappa-B signaling pathway, and TIR domain; dark blue—I-kappa-B kinase/NF-kappa-B signaling, and interleukin-1 receptor binding; yellow—TNFR1-induced NF-kappa-B signaling pathway, and TICAM1 deficiency—HSE; purple—innate immunity; green—cytokine; light blue—inflammatory responses; String software; detected genes showed no more than five interactions).

**Figure 5 viruses-13-02054-f005:**
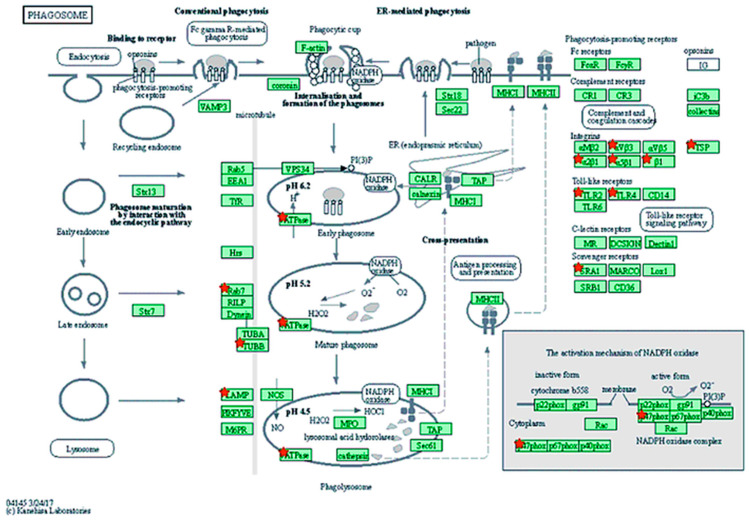
DEGs for which expression has been modified through SRLV infection involved in phagosome pathways (KEGG chx04145). The genes identified as differentially expressed (adjusted *p*-value < 0.05) between uninfected and infected goats were highlighted red.

**Figure 6 viruses-13-02054-f006:**
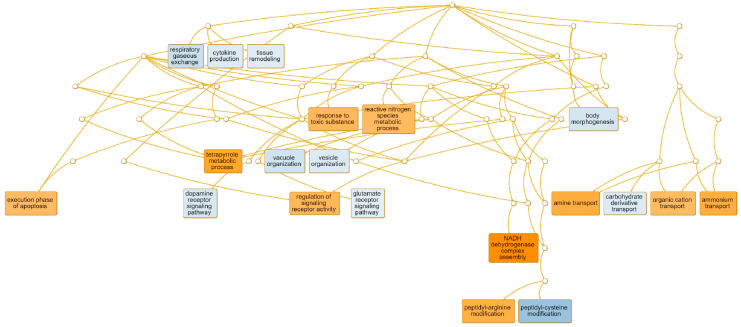
Tree graph represented the top 20 (10 up and 10 down) most regulated GO terms. Orange and blue color represented down and upregulated GO term, respectively. (WebGestalt, WEB-based GEne SeT AnaLysis Toolkit).

**Table 1 viruses-13-02054-t001:** The significant GO terms enrichment between uninfected and infected goats.

GO	Accession Number	Number of Genes	FDR	Identified Genes
MyD88-dependent toll-like receptor signaling pathway	GO: 0002755	4	0.019	*TLR2*, *TLR4*, *TLR8*, *TLR7*
Response to growth factor	GO: 0070848	5	0.050	*HHIP*, *BMPR1B*, *ADAMTS3*, *PDGFRB*, *BMP6*
Regulation of cytokine secretion	GO: 0001817	3	0.050	*TLR2*, *TLR4*, *TLR8*
Inflammatory response	GO: 0006954	4	0.050	*TLR2*, *TLR4*, *TLR8*, *TLR7*
Transmembrane signaling receptor activity	GO: 0004888	3	0.010	*TLR2*, *TLR4*, *TLR7*
Innate immune response	GO: 0045087	3	0.010	*TLR2*, *TLR4*, *TLR8*
Extracellular region	GO: 0045087	6	0.010	*KITLG*, *BMP6*, *INHBB*, *INSL3*, *IGFBP1*, *IGFBP3*
Integral component of membrane	GO: 0016021	13	0.010	*ATP6*, *KITLG*, *ND3*, *ND4*, *BMPR1B*, *CALCRL*, *COX1*, *DGAT2*, *SLC11A1*, *SCD*, *TLR2*, *TLR8*, *TLR7*

FDR—false discovery rate (*p*-value adjusted), GO—gene ontology.

**Table 2 viruses-13-02054-t002:** Significantly overrepresented pathways involved genes associated with SRLV infection.

Biological Pathways	Number of Genes Upregulated	Upregulated Genes	Number of Genes Downregulated	Downregulated Genes	FDR
Toll-like receptor signaling pathway	6	*LY96*, *PIK3R5*, *TLR2*, *TLR4*, *TLR7*, *TLR8*	1	*MAPK12*	0.050
Rheumatoid arthritis	4	*ATP6V1A*, *TLR2*, *TLR4*, *TGFB2*	0	*-*	0.010
Ras signaling pathway	5	*GAB2*, *EFNA4*, *KDR*, *PIK3R5*, *PDGFRB*	5	*KITLG*, *ANGPT2*, *PAK6*, *PLA2G1B*, *RGL1*	0.010
PI3K-Akt signaling pathway	8	*CSF3R*, *EFNA4*, *GYS1*, *KDR*, *PIK3R5*, *PDGFRB*, *TLR2*, *TLR4*	4	*KITLG*, *ANGPT2*, *COMP*, *COL1A1*	0.010
TNF signaling pathway	3	*TNFRSF1A*, *PIK3R5*, *SOCS3*	1	*MAPK12*	0.010
Phagosome	9	*ATP6V1A*, *RAB7B*, *LAMP2*, *MSR1*, *NCF1*, *TLR2*, *TLR4*, *TUBB1*, *TUBB4A*	1	*COMP*	0.020
Cytokine-cytokine receptor interaction	9	*TNFRSF1A*, *CSF2RB*, *CSF3R*, *CRLF2*, *IFNGR1*, *IFNGR2*, *IL1R2*, *PPBP*, *TGFB2*	5	*CCR10*, *TNFRSF13C*, *TNFRSF6B*, *AMH*, *BMPR1B*,	0.004

False discovery rate (*p*-value adjusted).

**Table 3 viruses-13-02054-t003:** The significant GO terms of downregulated genes in group HPL goats.

GO	Accession Number	Number of Genes	FDR	Identified Genes
execution phase of apoptosis	GO: 0097194	10	0.0240	*PTGIS*, *ENDOG*, *BOK*, *ERN2*, *SHARPIN*, *CIDEC*, *CXCR3*, *SIRT2*, *RPS3*, *BAX*
tetrapyrrole metabolic process	GO: 0033013	9	0.00679	*ALAS2*, *ALAD*, *UROD*, *CYP1A2*, *UROS*, *HNF1A*, *MMAB*, *ATP5IF1*, *BLVRB*
response to toxic substance	GO: 0009636	43	0.00691	*HBM*, *CHRNB2*, *LYPD1*, *GNRH1*, *MPO*, *CHRNA6*, *DRD3*, *CHRND*, *LTC4S*, *IL6*
regulation of signaling receptor activity	GO: 0010469	53	<0.0000	*FGF16*, *CCL2*, *CGA*, *INHBE*, *CLEC11A*, *CXCL5*, *TNFSF11*, *AVP*, *LYPD1*, *FOXH1*, *GNRH1*, *NPY*, *CCL17*, *OXT*, *GHRH*, *NOG*, *EDA*, *CXCL9*, *MIF*, *RETN*
peptidyl-arginine modification	GO: 0018195	10	0.01500	*ART1*, *PADI6*, *PADI3*, *PADI1*, *KRTCAP2*, *COPRS*, *PARK7*, *PRMT7*, *PRMT2*, *PRMT1*
amine transport	GO: 0015837	15	0.0060	*CHRNB2*, *CHRNA6*, *DRD3*, *ACE2*, *SNCG*, *SYT2*, *SLC22A16*, *SYT1*, *SLC18A1*, *DTNBP1*
organic cation transport	GO: 0015695	3	0.0290	*SLC22A1*, *SLC22A16*, *SLC18A3*
ammonium transport	GO: 0015696	15	<0.0000	*RHAG*, *CHRNB2*, *CHRNA6*, *DRD3*, *SLC6A2*, *SNCG*, *SYT2*, *ADCYAP1*, *SLC22A16*, *SYT1*
NADH dehydrogenase complex assembly	GO: 0010257	21	0.0010	*NDUFA13*, *NDUFS5*, *NDUFS7*, *NDUFB9*, *NDUFA9*, *NDUFA8*, *BCS1L*, *NDUFB2*, *NDUFA5*, *NDUFA2*

FDR—false discovery rate (*p*-value adjusted), GO—gene ontology, HPL—high proviral load.

**Table 4 viruses-13-02054-t004:** The significant GO terms of upregulated genes in group HPL goats.

GO	Accession Number	Number of Genes	FDR	Genes
peptidyl-cysteine modification	GO: 0018198	7	0.0020	*GOLGA7B*, *MAP6D1*, *TMX3*, *RAB3D*, *ZDHHC23*, *RAB6A*, *ZDHHC22*
cytokine production	GO: 0001816	95	<0.000	*ADAMTS3*, *IL15*, *CMKLR1*, *NOD2*, *ABCA1*, *GDF2*, *CCR2*, *HFE*, *RSAD2*, *CHUK*, *ITK*, *LBP*, *ADCY7*, *IL6R*, *IL1A*, *RAB7B*, *IFIH1*, *C3AR1*, *LPL*, *CD274*
vacuole organization	GO: 0007033	34	0.0351	*ABCA1*, *PIP4K2B*, *UBXN2A*, *RALB*, *RAB7B*, *PPT1*, *GAA*, *ACP2*, *TBC1D14*, *AKTIP*
vesicle organization	GO: 0016050	38	0.0300	*SEC16B*, *RAB8B*, *ABCA1*, *SAMD9*, *VPS39*, *RAB7B*, *DYSF*, *STX19*, *EXOC8*, *BCL2*
glutamate receptor signaling pathway	GO: 0007215	5	0.0600	*CACNG3*, *GRM2*, *CRHBP*, *HOMER2*, *PRNP*
tissue remodeling	GO: 0048771	14	0.0600	*IHH*, *DCSTAMP*, *RASSF2*, *CCR2*, *DLL4*, *RAB3D*, *IL1A*, *EFNA2*, *NOTCH2*, *CLDN18*

FDR—false discovery rate (*p*-value adjusted), GO—gene ontology, HPL—high proviral load.

**Table 5 viruses-13-02054-t005:** The selected genes in goats that were significantly upregulated with a high number of provirus copies (HPL) involved in multiple biological processes of cytokine production (GO: 0001816; FDR < 0.000).

Gene	Protein Name	FC	Adjpval	Protein Function
*TLR4*	Toll-like receptor 4	1.49	0.02	Acts via MYD88, TIRAP, and TRAF6, leading to NF-kappa-B activation, cytokine secretion, and the inflammatory response.
*TLR2*	Toll-like receptor 2	1.60	0.03	Related to mediating the innate immune response to bacterial lipoproteins or lipopeptides, related to cytokine secretion and the inflammatory response.
*TLR6*	Toll-like receptor 6	1.66	0.04	Acts via MYD88 and TRAF6, leading to NF-kappa-B activation, cytokine secretion, and the inflammatory response.
*CHUK*	Inhibitor of nuclear factor kappa-B kinase subunit alpha	2.44	0.03	Plays an essential role in the NF- kappa-B signaling pathway activated by multiple stimuli also by viral products.
*CSF1R*	Macrophage colony-stimulating factor 1 receptor	1.80	0.03	Controlling the proliferation and differentiation of hematopoietic precursor cells, especially mononuclear phagocytes, such as macrophages and monocytes.
*IRF1*	Interferon regulatory factor 1	1.41	0.02	Regulation of IFN and IFN-inducible genes, host response to viral and bacterial infections.
*NRLP3*	NACHT, LRR, and PYD domain-containing protein 3			Plays a crucial role in innate immunity and inflammation.
*IFIH1*	Interferon-induced helicase C domain-containing protein 1	2.14	0.05	Plays a major role in sensing viral infection and in the activation of a cascade of antiviral responses, including the induction of type I interferons and proinflammatory cytokines.
*TBK1*	Serine/threonine-protein kinase TBK1	1.80	0.02	Regulation of transcriptional activation of proinflammatory and antiviral genes including IFNA and IFNB.
*CD14*	Monocyte differentiation antigen CD14	1.67	0.05	Acts via MyD88, TIRAP, and TRAF6, leading to NF-kappa-B activation, cytokine secretion, and the inflammatory response.
*MYD88*	Myeloid differentiation primary response protein MyD88	1.82	0.01	Acts via toll-like receptor and IL-1 receptor signaling pathway in the innate immune response.

FC—fold change.

**Table 6 viruses-13-02054-t006:** Significantly overrepresented pathways involved genes for which expressions were associated with SRLV copy numbers.

Biological Pathways	Number of Genes Upregulated	Upregulated Genes	Number of Genes Downregulated	Downregulated Genes	adj*P* *
B-cell receptor signaling pathway	8	*LYN*, *CHUK*, *DAPP1*, *GRB2*, *PIK3CA*, *PIK3CB*, *PIK3AP1*, *PIK3R1*	4	*NFKBIB*, *HRAS*, *CD79B*, *CD79A*	0.018
Fc gamma R-mediated phagocytosis	7	*CRKL*, *LYN*, *ARPC5*, *PIK3CA*, *PIK3CB*, *PIK3R1*, *PRKCD*	5	*DOCK2*, *LAT*, *PLPP2*, *PRKCG*, *RPS6KB2*	0.034
Apoptosis	6	*FAS*, *CHUK*, *PIK3CA*, *PIK3CB*, *PIK3R1*, *TNFSF10*	4	*AIFM1*, *ENDOG*, *IL3RA*, *NTRK1*	0.034
Natural killer cell-mediated cytotoxicity	8	*FAS*, *GRB2*, *IFNAR1*, *PIK3CA*, *PIK3CB*, *PIK3R1*, *PTK2B*, *TNFSF10*	4	*HRAS*, *HCST*, *LAT*, *PRKCG*	0.057
T-cell receptor signaling pathway	7	*CHUK*, *GRB2*, *MAPK14*, *PIK3CA*, *PIK3CB*, *PIK3R1*, *TEC*	3	*HRAS*, *NFKBIB*, *LAT*	0.036
mTOR signaling pathway	5	*EIF4E*, *PIK3CA*, *PIK3CB*, *PIK3R1*, *ULK2*	2	*PRKCG*, *RPS6KB2*	0.037
FoxO signaling pathway	13	*EP300*, *CHUK*, *CDKN1A*, *GADD45A*, *GRB2*, *MAPK14*, *PIK3CA*, *PIK3CB*, *PIK3R1*, *PRKAB1*, *STAT3*, *TGFBR2*, *TNFSF10*	3	*HRAS*, *FOXO1*, *G6PC3*	0.034
TNF signaling pathway	9	*FAS*, *TNFRSF1B*, *CHUK*, *IL15*, *MAPK14*, *NOD2*, *PIK3CA*, *PIK3CB*, *PIK3R1*	0	*-*	0.044
Toll-like receptor signaling pathway	8	*TBK1*, *CHUK*, *IFNAR1*, *MAPK14*, *PIK3CA*, *PIK3CB*, *PIK3R1*, *TLR6*	0	*-*	0.044
Regulation of actin cytoskeleton	10	*CRKL*, *GIT1*, *IQGAP1*, *ARPC5*, *CYFIP1*, *PXN*, *PIK3CA*, *PIK3CB*, *PIK3R1*, *PPP1CB*	4	*HRAS*, *FGFR2*, *PFN1*, *PPP1CA*	0.017
HIF-1 signaling pathway	8	*EP300*, *CUL2*, *CDKN1A*, *EIF4E*, *PIK3CA*, *PIK3CB*, *PIK3R1*, *STAT3*	4	*TIMP1*, *FLT1*, *PRKCG*, *RPS6KB2*	0.054
Inflammatory mediator regulation of TRP channels	8	*ADCY7*, *MAPK14*, *PIK3CA*, *PIK3CB*, *PIK3R1*, *PLA2G4A*, *PRKCD*, *PPP1CB*	4	*CALM3*, *NTRK1*, *PRKCG*, *PPP1CA*	0.050
Signaling pathways regulating pluripotency of stem cells	8	*JAK1*, *JAK2*, *GRB2*, *MAPK14*, *PIK3CA*, *PIK3CB*, *PIK3R1*, *STAT3*	2	*HRAS*, *FGFR2*	0.049
Ras signaling pathway	11	*RAP1B*, *RALB*, *TBK1*, *CSF1R*, *CHUK*, *GRB2*, *PIK3CA*, *PIK3CB*, *PIK3R1*, *PLA2G4A*, *RALBP1*	6	*HRAS*, *CALM3*, *FGFR2*, *FLT1*, *LAT*, *PRKCG*	0.033

*P* * value adjusted using Benjamini–Hochberg correction.

**Table 7 viruses-13-02054-t007:** The correlation coefficients obtained for RNA-seq data validation using qPCR and between qPCR results and provirus copy number.

	Correlation Coefficient	
Gene Symbol	qPCR vs. RNA-seq ^1^	qPCR vs. Provirus Copy Number ^2^
*CCL2*	0.850	0.778 ***
*IL15*	0.919 **	−0.242 ^ns^
*CXCR3*	0.794 *	0.359 ^ns^
*MIF*	−0.248 ^ns^	−0.270 ^ns^
*NOD2*	0.433 *	−0.470 *
*CCR*	0.441 ^ns^	0.515 **
*BCL2*	−0.107 ^ns^	0.673 **
*CXCL5*	−0.232 ^ns^	0.759 ***
*ITK*	−0.589 ^ns^	0.478 *

^1^ correlation between qPCR and RNA-seq data; ^2^ correlation between qPCR data estimated for all goats tested by qPCR and provirus copy number; * *p*-value < 0.05; ** *p*-value < 0.01; *** *p*-value < 0.0001, −*p*-value < 0.1, ns—not significant.

## Data Availability

The data sets used and/or analyzed during the current study are available from the corresponding author on reasonable request.

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
