# Peer review of "Transcriptome Analysis for Genes Associated with Small Ruminant Lentiviruses Infection in Goats of Carpathian Breed"

_viruses, 2021, doi:10.3390/v13102054_

Round 1

Reviewer 1 Report

The manuscript has been modified according to the reviewer’s comments, I therefore suggest the publication.

Author Response

We would like thank the reviewer for his comment on our manuscript.

Reviewer 2 Report

This manuscript described the  Transcriptome analysis for genes associated with small ruminant lentiviruses infection in goats of Carpathian breed.

The methods and content of the work are clear.

However in the introduction other studies on resistence/susceptibility of goats to SRLV infection should be reported.

The possibility to manage SRLV infection and/or prevent the disease through genetic selection is  very important to preserve goats biodiversity (especially in endangered breeds)

Author Response

We would like thank the reviewer for his comments on our manuscript.  According to reviewer's comment  other studies on resistence/susceptibility of goats to SRLV infection have been added in the introduction.

This manuscript is a resubmission of an earlier submission. The following is a list of the peer review reports and author responses from that submission.

Round 1

Reviewer 1 Report

The authors descripe the changes in gene expression for goat with low and high viral load of SRLV. 

They investigated naturally infected goat from a flock with almost 100% seroprevalence. Although a natural infection mimics the disease outcome better than exprimental infection it also leave the study with a less controlled environment. No information can be given to at which time the animals were infected. Therefore I can't be said if the difference in viral load might be due to time of infection and partial clearance of the virus since infection. Samples should have been taken at different time points to check viral load over time to account for this factor. Furthermore, only animals with low and high viral load were compared to each other with no comparison to healthy animals. Genes uprealted by either group could still be down-regulated in comparison to healthy animals and results are miss interpreted this way.

Furthermore, the virus itself is not considered in any way in this experiment. Although sequencing is done no seuquencing results are shown for the virus. Different pro-viral loads might show mutation within the viral genome that might additionally acount for a higher viral load.

Although the technical side of this publication is without any major flawd the experimental design lacks sufficient controls to justify the conclusions drawn in the publication.

Reviewer 2 Report

The study reports the analysis of gene expression of blood cells of two groups of goats, naturally infected by SRLV, carrying high and low proviral loads. Differentially expressed genes, between the two groups, were identified by RNA sequencing and further confirmed by RT PCR for some of them.

The manuscript is interesting and well written; the experiments are clearly described and the results as well. I suggest the following modification to improve the quality of the article

  • The sentences at lines 83-89 describe the results of the study and their significance. They should be removed from introduction and placed in the conclusion session.
  • Line 219, Caprus hircus should be replaced by Capra hircus
  • A legend should be provided for supplementary table S2.
  • Format of references should be verified, in particular a unique criteria should be adopted to report the reference author lists
  • Figure 4, is not useful since it doesn’t provide new elements to the study, it may be moved to supplemental materials.